# Effect of Different Exercise Types on the Cross-Sectional Area and Lumbar Lordosis Angle in Patients with Flat Back Syndrome

**DOI:** 10.3390/ijerph182010923

**Published:** 2021-10-17

**Authors:** Won-Moon Kim, Yong-Gon Seo, Yun-Jin Park, Han-Su Cho, Chang-Hee Lee

**Affiliations:** 1Department of Sports Science, Dongguk University, 123, Dongdae-ro, Gyeongju-si 38066, Korea; kimwonmoon3426@hanmail.net; 2Division of Sports Medicine, Department of Orthopedic Surgery, Samsung Medical Center, 81, Irwon-ro, Gangnam-gu, Seoul 06351, Korea; 3Department of Health Rehabilitation, Osan University, 45, Cheonghak-ro, Osan-si 18119, Korea; africca3535@gmail.com; 4Sports Medicine Center, Sunsoochon Hospital, 76, Olympic-ro, Songpa-gu, Seoul 05556, Korea; suya16@hanmail.net; 5Department of Sports Science, Hanyang University, 55, Hanyang Daehak-ro, Sangnok-gu, Ansan-si 15588, Korea; world32001@naver.com

**Keywords:** spinal curvature, cross-sectional area, disability evaluation, exercise therapy

## Abstract

Flat back syndrome (FBS) is a sagittal imbalance wherein the normal spinal curvature is reduced. This study aimed to compare the effects of different exercise programs on the cross-sectional area (CSA) of the lumbar muscles, lumbar lordosis angle (LLA), lumbar disability, and flexibility in patients with FBS. Thirty-six females with flexible FBS were randomly allocated to the corrective exercise group (CEG, *n* = 12), resistance exercise group (REG, *n* = 12), and physical therapy group (PTG, *n* = 12). CEG and REG patients participated in a 12-week exercise intervention for 60 min three times per week. The CSA, LLA, Oswestry disability index (ODI), and sit-and-reach test were measured before and after intervention. CSA showed a significant difference between groups (*p* < 0.01), with CEG and REG demonstrating a significant increase (*p* < 0.05 and *p* < 0.05, respectively). LLA showed a significant difference between groups (*p* < 0.001); CEG showed a higher increase than did REG (*p* < 0.01) and PTG (*p* < 0.001). ODI also showed a significant difference between groups (*p* < 0.001), being lower in CEG than in REG (*p* < 0.001) and PTG (*p* < 0.001). Lumbar flexibility significantly improved in all groups, albeit with a significant difference (*p* < 0.001). Although corrective and resistance exercise programs effectively improve these parameters, corrective exercise is superior to other interventions for patients with FBS.

## 1. Introduction

Flat back syndrome (FBS) is a type of sagittal imbalance in the spine, and it is characterized by loss of the lordotic curve [1]. Although the pathogenic mechanism remains unclear, poor posture and lack of exercise are major causes of spinal deformity [2,3]. The normal curve of the spine is known to have a buffering effect against gravity and provides optimal, coordinated body movements [4].

An abnormal spinal curvature is associated with musculoskeletal and nervous systems problems [5]. Long-standing spinal imbalance is known to cause scoliosis, pelvic malalignment syndrome, tension-type headache, and nerve entrapment syndrome [6]. Therefore, it is very important to keep the optimal curve of the spine, and intensive intervention is required for abnormal alignment in patients with FBS.

Various interventions, including brace wearing, surgical correction, and exercise therapy, have been used to correct an abnormal spinal curvature. In the past, a surgical approach was mainly used to correct the spinal curve in patients with FBS [2]. The intervention for correcting the spinal curvature is influenced by various factors such as the patient’s age, sex, stage of spinal disease, and etiology of the disease [7].

Several studies have reported that exercise therapy as a nonsurgical approach can improve an abnormal spinal curvature in FBS [3,5,8]. A previous study [9] reported a significant improvement in the curvature of the cervical, thoracic, and lumbar vertebrae after exercise intervention. Another study conducted by Harrison and Oakley [1] reported a significant change in the spinal alignment and pain scale score after the lumbar renal traction method and manual therapy. However, most of the studies have focused on parameters such as the spinal curve, function, pain, and muscle strength, and few have compared different exercise programs for FBS.

Therefore, this study aimed to investigate the effects of different exercise programs on the cross-sectional area (CSA) of the lumbar muscle, the lumbar lordosis angle (LLA), lumbar function, and flexibility in patients with FBS.

## 2. Materials and Methods

### 2.1. Study Design

The G-power program (Heinrich Heine University Düsseldorf, Düsseldorf, Ger-many, version 3.1.9.4) [10] was used to determine the sample size using the effect size = 0.25, α = 0.05, power = 0.80%, the number of groups = 3, and the number of measurements = 2. The minimal sample size was calculated as 42. Therefore, 42 patients with diagnosed with FBS were enrolled, but six patients refused to participate due to personal reasons before starting this study. Finally, thirty-six female patients with flexible FBS participated in this study. The inclusion criteria were as follows: no improvement in symptoms including back pain for the past 3 months despite drug therapy, a thoracic kyphosis angle ≤30° of Cobb’s angle [9], and moderate disability with back pain (over 3 in numeric pain rating scale) according to the Oswestry disability index (ODI) (21–40%). Patients were excluded from the study if they had previously other spinal disease or had other musculoskeletal disease or had undergone spine surgery.

The participants were randomly assigned into a corrective exercise group (CEG, *n* = 12), a resistance exercise group (REG, *n* = 12), and a physical therapy group (PTG, *n* = 12) according to the order of enrollment in the study. All patients received sufficient explanations regarding the study and subsequently signed an informed consent form before participation. This study was conducted according to the guidelines of the Declaration of Helsinki, and all procedures involving patients were approved by the Ethics Committee of the relevant institution (IRB No. DGU-20200027).

### 2.2. Outcome Assessments

All assessments in this study were conducted before and after exercise intervention with the same method.

### 2.3. Cobb’s Angle

The LLA was evaluated using lateral radiographs (REX-52R; Listem, Donghwari, Korea) obtained with the patient standing. Cobb’s angle [11] was measured as the angle between the extended line along the upper border of L1 and the extended line along the lower border of L5 (Figure 1).

### 2.4. Cross-Sectional Area

The CSA of the lumbar muscle was measured using computed tomography (CT) (Sytec-Sri; GE, Boston, MA, USA). One radiologist evaluated all the images and measurements to ensure reliability. The patients laid in the supine position with the knee joint flexed at 25°–30°, and images were acquired for a total of 2 to 4 min with maintenance of a natural breathing pattern. CT parameters to acquire images of the CSA of the lower border of the fourth lumbar vertebral body were as follows: voltage, 120 kV (peak); current, 240 mA/s; thickness, 5 mm; and field of view, 35–40 cm. On the monitor of the picture archiving and communication system, the CSA (cm^2^) of the paraspinal lumbar muscles including longissimus, iliocostalis, and multifidus was analyzed and calculated (Figure 2).

### 2.5. Oswestry Disability Index

The ODI is a scale designed to measure disability in daily life caused by lower back pain. The scale consists of 10 items, including pain intensity, personal care, lifting objects, sitting, walking, sleeping, standing, sexual activity, travel, and social life. Each item is scored from 0 to 5, with 0 representing no discomfort and 5 representing the most severe discomfort. A higher score represents more severe disability [12].

### 2.6. Sit-and-Reach Test

The sit-and-reach test was conducted using the DWR-OT1038 device (Dawoori, Anseong-si, Korea). The patients sat with their legs together, knees extended, and upper body erect. With their hands placed side by side or overlapping, and while exhaling and bending their upper body forward, the patients reached forward with their arms as far as possible and pushed an indicator. The participant was instructed to maintain the farthest position for 3 s, and the examiner recorded the position of the indicator at this point [13].

### 2.7. Exercise Programs

In this study, corrective and resistance exercises were performed by two groups, whereas the third group received only physical therapy. The corrective exercise program was based on the principles of the Schroth exercises and focused on correction of thoracolumbar spine. The program was revised by referring to the findings of previous studies [14,15]. The exercise program was conducted for 60 min three times per week for 12 weeks. The exercise intensity was prescribed according to a rating of perceived exertion (RPE) of 13 with “somewhat hard” to 15 with “hard”. The resting time between each exercise set was 60 s. The detailed corrective exercise program is summarized in Table 1.

The resistance exercise program was consisted of trunk exercise and whole body muscle strength exercise, and the whole body exercise was performed with elastic resistance bands (Performance Health, Akron, OH, USA). This program was modified from previous studies [16,17]. The initial exercise intensity in elastic resistance bands was set at 30–40% of one-repetition maximum and was progressively increased up to 60–70% during the intervention. The resting time between each set was 60 s. Details of the resistance exercise program are summarized in Table 2.

Physical therapy was conducted by a physical therapist for 60 min. It included 20 min of ultrasound therapy at a frequency of 1 MHz and an intensity of 1.6 W/cm^2^, 20 min of interferential current therapy, and 20 min of hot pack application in the thoracolumbar spine region.

### 2.8. Statistical Analysis

The Kolmogorov–Smirnov test was used to confirm normal distribution of the data. Means and standard deviations were used for descriptive statistics of all parameters. One-way analysis of variance was used to determine the differences in clinical outcomes among the three groups. Post-hoc analysis was performed when there was a significant difference between groups. All statistical analyses were performed using SPSS Statistics version 22.0 (IBM, Armonk, NY, USA), and the level of significance was set at *p* < 0.05.

## 3. Results

Thirty-six patients with FBS participated in this study, and there were no significant differences in the baseline characteristics of the three groups (Table 3).

With regard to the CSA of the lumbar muscle, there was a significant difference among the three groups (*p* < 0.01). Post hoc analysis showed a significant increase in the CEG and REG, relative to that in the PTG (*p* < 0.05, *p* < 0.05, respectively). The LLA also showed a significant difference among groups (*p* < 0.001), with a significantly higher value in the CEG than in the REG (*p* < 0.01) and PTG (*p* < 0.001). In addition, the REG showed a more significant improvement than did the PTG (*p* < 0.001).

The ODI was also significantly different among groups (*p* < 0.001). Post hoc analysis showed that the ODI was significantly lower in the CEG than in the REG (*p* < 0.001) and PTG (*p* < 0.001), while it was significantly lower in the REG than in the PTG (*p* < 0.001).

Changes in flexibility were significantly different among groups (*p* < 0.001). Post hoc analysis revealed that the CEG and REG showed a more significant increase than did the PTG (*p* < 0.001, *p* < 0.001, respectively). There were significant improvements in the CSA, the LLA, the ODI, and flexibility after the interventions in the CEG (*p* < 0.001) and REG (*p* < 0.001), whereas there was no significant improvement in any parameter except flexibility (*p* < 0.05) in the PTG (CSA, *p =* 0.725; LLA, *p =* 0.491; ODI, *p =* 0.136, respectively; Table 4).

## 4. Discussion

Flat back syndrome is associated with malalignment of the spinal curve, resulting in a forward head posture and lower back pain [18]. This study compared the effects of different exercise types on the CSA, the LLA, the ODI, and flexibility of the spine and found that compared with physical therapy, both corrective and resistance exercises are effective in improving these parameters in patients with FBS.

The deep lumbar muscles play an important role in maintaining control and stability of the spinal column [19]. The presence of a flat back is associated with malalignment in the spine, which could cause dysfunction of the deep lumbar muscles and result in chronic low back pain and deep muscle atrophy [20,21]. In the present study, the CSA of the lumbar muscles showed a more significant increase in the CEG and REG than in the PTG. This result is similar to the findings of Cho et al. [22], who demonstrated that lumbar extension exercises can increase the CSA of the deep paraspinal muscles, including the longissimus, iliocostalis, spinalis, and multifidus. This suggests that improvement of the lordotic curve in the lumbar region after exercise intervention contributes to a decrease in the overload on the lumbar vertebrae and increases the lumbar muscle activity, resulting in an increase in the CSA of the lumbar muscles. This mechanism is supported by previous studies reporting the positive effects of the Schroth and mobilization exercises on muscle activity [19,23]. Further study is needed to confirm whether corrective exercise or resistance exercise would be more beneficial in terms of an increase in the CSA of the lumbar muscle in patients with FBS.

The ideal curvature of the spine in the sagittal plane serves to reduce loads on the vertebral discs and any shock to the spine, and it allows effective action of the spinal muscles [24]. An abnormal LLA caused by thoracolumbar kyphosis can have negative effects on the overall biomechanics of the spine [25]. Decreased thoracolumbar motion is associated with the range of motion of the lumbar spine, and it can reduce lumbar lordosis [26]. In the present study, corrective and resistance exercises resulted in greater improvements in the LLA than did physical therapy alone. However, the corrective exercise program was the most effective in improving lumbar lordosis. Several studies have reported that corrective exercise programs improved the LLA [22,27]. Particularly, several exercise modes, including mobilization, lumbar stabilization exercise, and the Schroth exercises are beneficial interventions to improve the LLA [23,28]. We speculate that the corrective exercises applied in this study, which comprised mobilization and the Schroth exercises, may have improved the LLA in patients with FBS.

Several studies [28,29,30] have reported that lumbar stabilization exercises reduce lumbar functional disability measured using the ODI in patients with chronic low back pain. In this study, the CEG and REG showed a lower ODI score than did the PTG, and the corrective exercise program, which included lumbar stabilization exercises, was superior to the other interventions for patients with FBS. One previous study [31] reported that flat back syndrome can be associated with functional limitations after surgical intervention. Most previous studies on lumbar functional disability have focused on patients with chronic low back pain or flat back who undergo surgery. To the best of our knowledge, this is the first study evaluating functional disability in patients with functional flat back. Further studies with larger samples are required to confirm our findings.

In a previous study, hamstring flexibility was significantly lesser in patients with low back pain than in healthy patients [32]. Tightness of the hamstring muscle is associated with the lumbar spinal curve, and it results in loss of the lordotic curve in the lumbar region [33,34]. Our results indicated that exercise interventions and physical therapy improved the flexibility of the hamstring and erector spinae muscles in patients with FBS, with the CEG and REG showing a greater increase in flexibility. Thus, corrective and resistance exercises could be beneficial in terms of improved flexibility in patients with FBS. Further research can confirm the mechanism underlying the improved flexibility.

This study has some limitations. First, all the participants were female, and our results may not be applicable to all populations, particularly because of differences in the prevalence of FBS between women and men. Therefore, further studies should confirm differences in these results between male and female patients. Second, the sample size was small, even though the patients were divided into three groups. A larger sample is necessary to validate our findings.

## 5. Conclusions

Corrective and resistance exercises are both effective in improving the CSA of the lumbar muscles, the LLA, the ODI, and flexibility, with better effects than those of simple physical therapy. However, corrective exercise programs seem to be the most appropriate intervention for patients with FBS.

## Figures and Tables

**Figure 1 ijerph-18-10923-f001:**
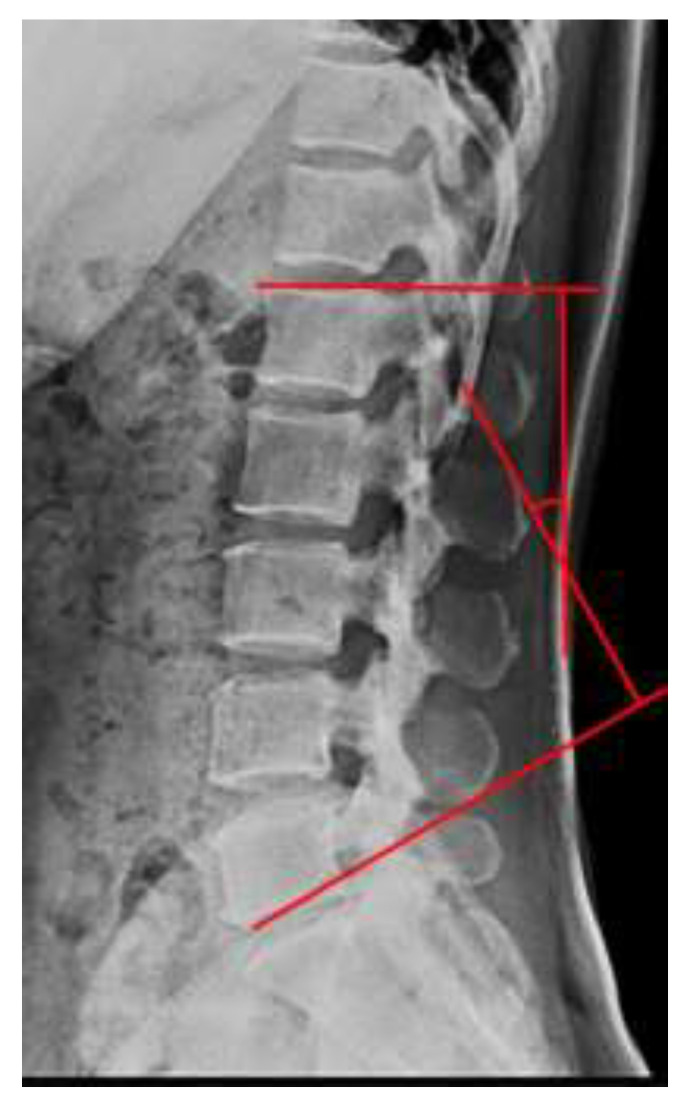
X-ray measurement of the lumbar lordosis angle (Cobb’s method).

**Figure 2 ijerph-18-10923-f002:**
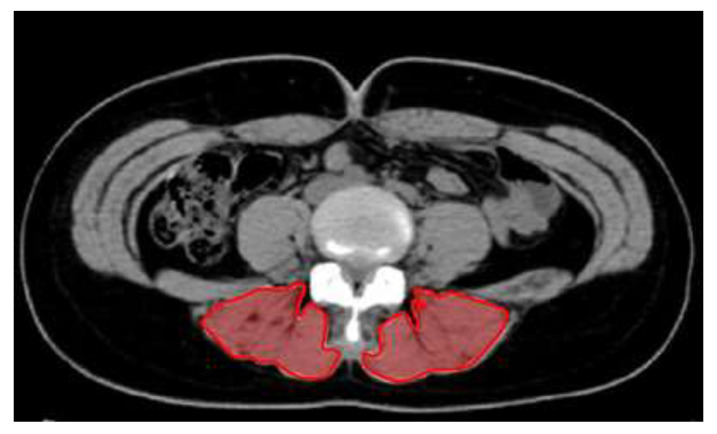
Computed tomography measurement of the cross-sectional area of the paraspinal lumbar muscle.

**Table 1 ijerph-18-10923-t001:** The corrective exercise program.

Corrective Exercise Program			
Exercise Type	Exercise Mode	Time	Intensity
Warm-up	Stretching for upper and lower body	5 min	RPE (10–13)
Mobilization exercise for correction	Anterior/posterior pelvis exercises using the Schroth breathing pattern	30 min	RPE (13–15) 15–20 Reps 3 Sets
	Thoracolumbar spine mobilization exercises using the Schroth breathing pattern		
	Lumbosacral spine mobilization exercises using the Schroth breathing pattern		
	Thoracic kyphosis mobilization exercises using the Schroth breathing pattern		
	Thoracolumbar lordosis mobilization exercises using the Schroth breathing pattern		
Corrective exercise for the thoracolumbar spine	Thoracolumbar corrective exercises using an exercise ball	30 min	RPE (13–15) 15–20 Reps 3 Sets
	Thoracolumbar corrective exercises using Pilates rings		
	Thoracolumbar corrective exercises using dumbbells		
	Thoracolumbar corrective exercises using tubing		
	Thoracolumbar corrective exercises using slings		
Cool-down	Stretching for upper and lower body	5 min	RPE (10–13)

RPE, rating of perceived exertion; Reps, repetitions.

**Table 2 ijerph-18-10923-t002:** The resistance exercise program.

Resistance Exercise Program			
Exercise Type	Exercise Mode	Time	Intensity
Warm-up	Stretching for upper and lower body	5 min	RPE (10–13)
Trunk exercise	Planks to strengthen the trunk muscles	30 min	RPE (13–15) 15–20 Reps 3 Sets
	Side planks to strengthen the trunk muscles		
	Functional planks to strengthen the trunk muscles		
Upper/lower body muscle-strengthen exercise with elastic resistance bands	Scapular retraction exercise	40 min	1RM of 30%–40% to 60–70% 15–20 Reps 3 Sets
	Push-up plus exercise		
	Lat pull-down		
	Squats		
	Lunges		
	Step-ups		
Cool-down	Stretching of upper and lower body	5 min	RPE (10–13)

RM, repetition maximum; RPE, rating of perceived exertion; Reps, repetitions.

**Table 3 ijerph-18-10923-t003:** Baseline characteristics of participants.

Characteristics	CEG	REG	PTG	*p*-Value
Numbers	12	12	12	-
Age (years)	38.83 ± 3.49	39.67 ± 2.84	39.83 ± 3.07	0.078
Height (cm)	159.03 ± 3.42	161.99 ± 3.29	161.77 ± 3.79	0.085
Weight (kg)	63.40 ± 6.11	61.89 ± 4.19	61.05 ± 4.84	0.528
BMI (kg/m^2^)	23.60 ± 2.14	23.58 ± 1.72	23.59 ± 1.75	0.999

Data are presented as mean ± standard deviation. CEG, corrective exercise group; REG, resistance exercise group; PTG, physical therapy group; BMI, body mass index.

**Table 4 ijerph-18-10923-t004:** Changes in clinical outcomes after the interventions.

Variable	Time	CEG	REG	PTG	F	Post Hoc
CSA (cm^2^)	Pre	20.30 ± 4.74	15.55 ± 2.53	19.83 ± 3.45	5.519 **	a > c * b > c *
	Post	24.53 ± 4.34 ^†††^	23.78 ± 2.49 ^†††^	19.96 ± 3.75		
LLA (°)	Pre	33.17 ± 1.85	33.50 ± 1.62	32.75 ± 2.09	32.960 ***	a > c *** a > b ** b > c ***
	Post	40.25 ± 2.73 ^†††^	37.17 ± 1.95 ^†††^	32.5 ± 2.32		
ODI	Pre	25.00 ± 1.81	23.58 ± 1.93	24.17 ± 1.53	31.788 ***	a > c *** a > b *** b > c *
	Post	14.58 ± 3.68 ^†††^	20.67 ± 2.45 ^†††^	23.50 ± 1.98		
Flexibility (cm)	Pre	0.67 ± 6.73	−0.17 ± 6.81	1.50 ± 4.96	28.997 ***	a > c *** b > c ***
	Post	14.25 ± 3.60 ^†††^	14.92 ± 4.60 ^†††^	2.50 ± 5.14 ^†^		

Values are presented as mean ± standard deviation. CEG, corrective exercise group; REG, resistance exercise group; PTG, physical therapy group; CSA, cross-sectional area; LLA, lumbar lordosis angle; ODI, Oswestry disability index; a, CEG; b, REG; c, PTG; ^†^
*p* < 0.05, ^†††^
*p* < 0.001: paired t test; * *p* < 0.05, ** *p* < 0.01, *** *p* < 0.001: one-way analysis of variance.

## Data Availability

No new data were created or analyzed in this study. Data sharing is not applicable to this article.

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
