# Peer review of "Effect of Different Exercise Types on the Cross-Sectional Area and Lumbar Lordosis Angle in Patients with Flat Back Syndrome"

_ijerph, 2021, doi:10.3390/ijerph182010923_

Round 1
Reviewer 1 Report
I am happy to have the opportunity to review your manuscript.
I understand that the aim of this study is to investigate the effect of different exercise types on Cobb’s angle, cross-sectional area of lumbar muscle and function in patients with flat back syndrome (FBS). The results demonstrated that corrective and resistance exercise were more effective than physical therapy, and that corrective exercise was more effective than resistance exercise in some results. The results in this study are interesting to me and it will be helpful in clinical application. However, there are some things to consider for improving the quality of this manuscript. Therefore, I suggest some comments for your study.
Major revision
- I know that flat back syndrome has various causes including iatrogenic, degenerative/postural, traumatically induced fractures, and insidious onset. Additionally, in a previous study conducted by Harrison (2018), FBS was categorized into two broad types, fixed and flexible. This study described the inclusion criteria for your study in line 69-72. Generally, we think that exercise therapy is a beneficial intervention on improvement of spinal curvature, lumbar function in flexible flat back but it would be not applied in patients with fixed flat back spine. Although the inclusion criteria for subject were referred from previous study [9], more detailed description to inclusion criteria is needed in study design session.
- What is the difference between corrective and resistance exercise in your study? I think that the type of applied exercise in this study is considered to be a very important variable by interpreting the result. The reason is that unlike other previous studies, the purpose of this study is to investigate effects of different exercise types in patients with FBS. The both exercise program was revised by referring to the findings of previous studies but more exact information should be provided about exercise programs for the reader. For example, what means the perceived exertion of 13 to 15 or explain the purpose of the exercise program.
Minor revision
- In line 73, author described the participants were randomly assigned into the three group. Please describe accurately the allocation method for this study.
- This study was checked lumbar lordotic angle using X-ray with Cobb’s angle method. Please insert the reference in line 86.
- In this study, cross-sectional area of lumbar muscle was measured using CT and provided the information on figure 2. Please provide the detail information of included paraspinal lumbar muscles measured for this study.
4. I don’t know whether this manuscript has been edited in English. I understand how difficult this can be, especially if English is not your first language, but for readers to understand and benefit from your manuscript the use of English must be held to a very high standard. I think that the proofreading English needs for helping understanding of the reader
Reviewer 2 Report
Thank you for the opportunity to review the article entitled Effect of different exercise types on cross-sectional area and lumbar lordosis angle in patients with the flat back syndrome
The authors discuss an important and very current problem of the loss of lordosis of the lumbar spine.
The flatback syndrome is often a pathology of the spine associated with a sedentary lifestyle and pain. It is clinically important to investigate interventions and procedures that rebuild the biomechanics of the spine.
After analyzing the text, the following questions and comments arise:
1. Where did the idea to compare the effects of exercises with physical therapy comes from? The analyzed endpoints are the angle of lordosis and the mass of deep muscles in the lumbar region. Physical therapy cannot influence these parameters in any theoretical way. Such comparative group planning does not make much sense.
2. The selection criteria should be given in the text, not just quoted,
It is not known whether these patients are with pain or other pathology of the spine.
3. The figure describing the methodology of the Cobb angle measurement - shows an x-ray image with L4 / L5 spondylolisthesis, such patients should not practice exercises increasing lordosis. The flattened angle of lordosis is the result of this pathology. Movements in direction of lordosis, increase the degree of slippage. Even if it is just a random photo, it makes a bad impression in comparison with the title and topic of the work.
Round 2
Reviewer 1 Report
The authors have done a nice job of revising this manuscript and addressing the reviewer comments and queries. The manuscript now reads with greater focus and clarity.